# Assessing Adolescent Dating Violence in the YourLife Project: Proposal of an Instrument for Spanish-Speaking Countries

**DOI:** 10.3390/ijerph18136824

**Published:** 2021-06-25

**Authors:** Cristina Lopez-del Burgo, Alfonso Osorio, Pedro-Antonio de la Rosa, María Calatrava, Jokin de Irala

**Affiliations:** 1Department of Preventive Medicine and Public Health, University of Navarra, 31008 Pamplona, Spain; cldelburgo@unav.es (C.L.-d.B.); jdeirala@unav.es (J.d.I.); 2Institute for Culture and Society (ICS), University of Navarra, 31009 Pamplona, Spain; pdelarosa@alumni.unav.es (P.-A.d.l.R.); mcalatrava@unav.es (M.C.); 3Navarra Institute for Health Research (IdiSNA), 31008 Pamplona, Spain; 4School of Education and Psychology, University of Navarra, 31009 Pamplona, Spain; 5Human Flourishing Program, Institute for Quantitative Social Science, Harvard University, Cambridge, MA 02138, USA

**Keywords:** adolescence, dating violence, validation, questionnaire

## Abstract

Background: Several instruments have been developed to assess adolescent dating violence but only few have been validated in Spanish-speaking settings. Some instruments are too long and may not be feasible to include them in a multipurpose questionnaire. We developed an instrument to be used in the YourLife project, an international project about young people lifestyles. Objective: We aimed to analyze the psychometric properties of this instrument in three Spanish-speaking countries (Chile, Ecuador, and Spain). Method: We included 1049 participants, aged 13–18 years. Exploratory and confirmatory factor analyses were conducted. Associations between dating violence and variables expected to covariate with it (substance use, school peer aggression, justification of dating violence, and relationship power imbalance), were tested. Results: Two different constructs (psychological and physical/sexual) for suffered and perpetrated violence were identified and confirmed in the three countries. The dating violence subscales had Cronbach’s alpha scores higher than 0.85. The strongest associations between dating violence and variables related to it were found within the relationship power imbalance items, suggesting that these items may be useful to detect adolescent dating violence when a specific questionnaire cannot be implemented. Conclusion: This instrument seems to be adequate to assess suffered and perpetrated adolescent dating violence within a multipurpose questionnaire among schooled adolescents.

## 1. Introduction

Adolescent dating violence (ADV) is defined as physical, sexual, or psychological/emotional abuse, including threats, towards the partner with whom one has a romantic relationship during adolescence. It can take place in person or virtually [1].

ADV is a serious public health problem. According to a systematic review of 113 studies, psychological dating violence ranges from 4.2% to 97% [2]. The majority of the studies included in this review were conducted in the USA, although studies from Canada, Spain, and several Scandinavian and Latin American countries were also reviewed. This great variability of dating violence prevalence is due to the different samples and instruments used by the researchers. Another meta-analytic review of 101 studies, the majority of them from the USA, found that 1 in 5 adolescents reported suffering physical ADV and 1 in 10 reported being victims of sexual ADV [3].

According to studies from the USA and European countries, psychological ADV is more frequent than physical or sexual violence [4,5,6]. Several studies, conducted in Spain and in the USA, also find a peak in physical partner violence at the age of 16–17 [7,8,9]. In addition, studies show that ADV can be perpetrated and suffered by both girls and boys, but some differences are found. Prevalence of perpetration of sexual and physical ADV is higher among boys, while psychological ADV is perpetrated more commonly by girls [2]. Bidirectional aggression (members of the couple being both perpetrators and victims in that same romantic relationship) is common among adolescents [4,10,11,12,13].

During adolescence, personal beliefs, such as gender stereotypes, myths about love, or justification of violence in relationships, and several behaviors, such as alcohol consumption or bullying, can lead to dating violence [2,14]. The link between alcohol and intimate partner violence is complex. Apart from the psychopharmacological effect of alcohol, other personal factors, such as antisocial personality traits, might explain the association between alcohol and violence [15]. Studies about bullying and dating violence show that both types of violence share risk factors such as low empathy or high impulsivity, among others [16].

Regarding social factors, reviews about determinants of ADV conclude that school bonding can be a protective factor for dating violence, but no data about the type of school (co-educational/single-sex or public/private) were available. Inconsistent results were found for socioeconomic status and residence area (rural/urban) [14,17].

The consequences of ADV for adolescent health are known. Apart from physical injuries, or even death, ADV has been associated with an increased risk of mental health disorders, depression, anxiety, eating disorders, suicidal behavior, and antisocial behaviors or other risk behaviors, such as abuse of alcohol and other drugs [18,19,20,21]. In addition, psychological violence can be a predictor of physical violence and adult intimate partner violence in the future [22]. According to a study conducted in a sample of engaged couples from Mexico, perceived relationship power imbalance seems to be a valid proxy of intimate partner violence and could lead to greater forms of violence later in marriage [23]. This issue has not been evaluated in dating adolescents.

These data highlight the importance of preventing dating violence during adolescence.

Several instruments have been developed to assess ADV, but only some of them exhibit adequate psychometric properties or have been validated in Spanish-speaking settings [24]. These include the Revised Conflict Tactic Scale (CTS2) [25], the Conflict in Adolescent Dating Relationship Inventory (CADRI) [26], the Questionnaire of Violence Partners (DVQ or, in Spanish, CUVINO) [27] and the Violence in Adolescents’ Dating Relationships Inventory (VADRI) [28]. The CTS2 and the CADRI do not include items to evaluate digital violence, a prevalent way of exerting violence among adolescents nowadays [29,30], and the CUVINO only assesses victimization but not perpetration of ADV. The VADRI has 52 items (26 items in both forms of victimization and perpetration), including psychological, sexual and physical violence items, which may make its implementation difficult in some cases.

The YourLife (YL) Project is an international ongoing research project about young people’s lifestyles and personal relationships [31]. It includes cross-sectional analyses and a longitudinal follow-up. Schooled adolescents aged from 12 to 17 years fill out a project-specific questionnaire. The initial YL questionnaire included, among other topics related to adolescent health, several items about psychological dating violence. Preliminary data from Spain, Chile, Peru, and Mexico, collected among November 2016 and October 2018, showed that psychological dating violence was indeed a frequent problem among adolescents from all the participant countries, especially among boys [32]. In light of those results, we proposed to further assess the problem of ADV among schooled adolescents. To achieve this goal, we decided to include new questions related to psychological, physical, and sexual ADV in the YL questionnaire. In many cases, a long instrument is not feasible, for example, when it is implemented together with other instruments or questions, and when there is limited time to fill it out, as is the case in the YL questionnaire. For this reason, we developed a new instrument, based on scientific bibliography and other instruments about ADV.

This study aimed to analyze the psychometric properties of the ADV questionnaire included in the YL project (named ADV-YL) using a sample of three Spanish-speaking countries. Although the cultural context may influence the prevalence and determinants of ADV in each country, no differences are expected in the structure of the instrument, as we are using a wide range of possible determinants.

## 2. Method

### 2.1. Sample

The YL sample included a total of 2254 participants from Chile (N = 727: 32.3%), Ecuador (N = 305: 13.5%), and Spain (N = 1222: 54.2%), aged between 13 and 18 years. For this study, we only used data from participants who reported having a partner or having had one in the past (N = 1140). We also dropped participants with one or more missing answers in the dating violence section in the questionnaire (N = 91). Therefore, the final sample size for this analysis was 1049 (mean age of the whole sample is 15.9 years, SD = 1.2). Around half (54%) were girls (mean age = 15.8 years, SD = 1.2), and 46% were boys (mean age = 16.0 years, SD = 1.2). The majority came from co-educational schools (88.7%) and from urban areas (92%).

### 2.2. Instrument and Variables

The items of the questionnaire, used for this analysis, can be classified into the following areas:

#### 2.2.1. Demographics

These questions collected information regarding sex (male/female), age (years), type of school (single-sex/co-educational), and location of school (urban/rural).

#### 2.2.2. Adolescent Dating Violence

The research team elaborated two scales about suffered and perpetrated ADV, taking into account the existing questionnaires and bibliography about ADV. Both scales encompassed eighteen items for psychological, physical, and sexual or violence. Digital violence was also included. Items can be found in Table 1. All possible responses to these items were formulated through a Likert Scale with a range between 0 (never) and 6 (very frequently).

#### 2.2.3. Relationship Power Imbalance

The questionnaire also included three items regarding relationship power imbalance: “I have felt frightened/afraid of my partner”, “I have felt trapped and unable to leave the relationship”, and “I have felt controlled, lacking freedom”. All possible responses to these items were formulated through a Likert Scale with a range between 0 (never) and 6 (very frequently). These variables were dichotomized into never (answer 0) versus ever (answers 1–6).

#### 2.2.4. School Peer Aggression

Participants were asked if they had suffered or perpetrated “some sort of physical or psychological aggression or harm (insulted, hit, ignored, etc.) by/to someone in their school”. Responses ranked from 0 = never to 6 = very frequently. Bullying is considered a repeated and deliberate act to hurt a specific person [33]. In our sample, students referring frequent aggression at school were less than 10%. Therefore, we decided to measure “any school aggression” and dichotomized these variables into never (answer 0) versus ever (answers 1–6).

#### 2.2.5. Risky Behaviors

Frequency of alcohol consumption, binge drinking (consumption of 5 or more alcoholic drinks within a 2 h period), cannabis, and other substance consumption was recorded from 0 = never to 4 = three days per week or more. For analysis purposes, these variables were dichotomized into never (scale 0) and ever (scales 1–4), as any substance use during adolescence entails health risks [34].

#### 2.2.6. Opinions about Violence in Romantic Relationships

Participants were asked about some beliefs regarding violence in romantic relationships: “Sometimes it is justifiable to hit your partner if they do something annoying”, “Sometimes it is justifiable to have sexual relations although the other person says they do not want to (for example, if they’ve started petting or kissing you)”, and “Sometimes it is justifiable to have sexual relations when the other person is too drunk/drugged to decide if they want to or not (for example, if they’ve started petting or kissing you)”. Responses were recorded with a Likert scale from 0 = completely disagree to 6 = completely agree. These variables were dichotomized into completely disagree (answer 0) versus any degree of agreement (answers 1–6).

### 2.3. Procedure

The research team invited high schools to participate in the study during the years 2019 and 2020. Ninety five percent of the sample responded to the questionnaire between June 2019 and February 2020, before COVID-19 pandemic was declared by the WHO. Only 5% responded during the pandemic (between November 2020 and February 2021). These participants were from Spain. In that period in Spain, there were some mobility restrictions but no lockdowns, and the students were attending schools in-person. As explained before, we only included participants who had a current partner or had a partner in the past. Among them, only 3.3% (n = 35) answered the questionnaire during the pandemic.

Schools’ e-mail addresses were obtained from official registers. All data were gathered through an online, self-administered, anonymous questionnaire during class time. Teachers and students were informed about the anonymity and confidentiality of the study. Students were free to leave the classroom at any given moment. As is usual in epidemiological surveys, they were informed that by agreeing to fill in the questionnaire, they were giving their consent. School staff were entrusted with the management of parental permission.

Ethical approval for the whole project was obtained from the Ethics Committee of the University of Navarra, Spain (project ref. 2018/077).

### 2.4. Analyses

Analyses were performed using STATA/SE version 15.0 [35]; we used two-sided *p*-values and the statistical significance threshold was set a priori at 0.05.

A Kaiser–Meyen–Olking measure of sampling adequacy and Bartlett’s test of sphericity were assessed for both suffered and perpetrated violence scales, in order to test whether the data were suitable to conduct exploratory factor analysis (EFA). The factor structure of both violence scales was then assessed through a principal factor analysis, with oblique Promax rotation [36]. The final number of principal factors was decided based on the minimum average partial correlation method, also considering the conceptual classification of the items. Items that showed an overlap in more than one factor were dropped from the respective subscale. After a final structure was chosen with the whole sample through EFA, we performed confirmatory factor analysis (CFA) within each country, in order to test whether the same structure was appropriate for the three countries [37]. Lastly, to examine the internal consistency, Cronbach’s alpha and Composite Reliability (CR) values were assessed for each subscale [38].

After defining the ADV subscales, we performed analyses to test if several variables were associated with ADV. Student’s *t* tests were performed to assess the difference of score means for all the ADV subscales across variable categories. Additionally, Cohen’s *d* values were estimated to evaluate the effect size between ADV scales and other variables often found to be associated with ADV.

Lastly, to assess if the associations between other variables and subscales were different within sex categories, all analysis were replicated stratifying by sex.

## 3. Results

Preliminary analyses showed a Kaiser–Meyer–Olkin of sampling adequacy of 0.937 and 0.947 for items of the suffered and perpetrated partner violence scales, respectively. Additionally, the *p* values of both Bartlett’s sphericity tests were <0.001, thus meaning that enough common variability exists between items to perform exploratory factorial analysis.

The minimum average method suggested taking two factors per subscale. Results from EFA are shown in Table 1. After EFA, two factors were offered from both suffered and perpetrated violence, with seven items in Factor 1 and 11 items in Factor 2. Both factors were tagged by authors as “psychological” violence and “physical and sexual” violence. Item 5 (“They threaten to hurt you if you leave them”) was designed to belong to psychological violence, but we found it saturated in the physical/sexual violence factor in both perpetrated and suffered violence scales. Then, a closer examination of the item showed that it had both psychological and physical qualities, so the item was dropped from the final subscales. Thus, four different ADV subscales were defined: (1) Suffered Psychological Violence, (2) Suffered Physical/Sexual Violence, (3) Perpetrated Psychological Violence and (4) Perpetrated Physical/Sexual Violence.

The structure suggested by EFA was then tested in CFA for each country. Table 2 shows the results for suffered and perpetrated violence in each country. Fit indices were very good in all cases, suggesting configural invariance (or construct invariance) across countries.

The four ADV scales had Cronbach’s alpha scores higher than 0.85 (suffered psychological = 0.865, suffered physical/sexual = 0.920, perpetrated psychological = 0.876, and perpetrated physical/sexual = 0.972) and CRs higher than 0.83 (suffered psychological = 0.854, suffered physical/sexual = 0.905, perpetrated psychological = 0.833, and perpetrated physical/sexual = 0.963). When internal consistency was measured within each country, all Cronbach alphas were higher than 0.83. All scales were positively correlated to each other (Figure 1). The correlations between subscales were moderate, except for perpetrated psychological ADV and perpetrated physical/sexual ADV, where correlation was strong.

Table 3 shows the associations between ADV subscales and several variables often found associated to ADV in the literature [2,14]. Binge drinking, cannabis, and other substance use were associated with higher scores in the four ADV subscales (*p* < 0.05). The largest differences were found between students that had ever consumed drugs and those that had never consumed (Cohen’s *d* > 0.40, *p* < 0.001). Alcohol consumption (excluding binge drinking) was only associated with higher scores in the suffered psychological subscale (Cohen’s *d* = 0.14, *p* < 0.01). As expected, all items regarding opinions about violence in relationships and relationship power imbalance were strongly associated with the four ADV subscales (*p* < 0.001). Being a victim of school peer aggression was associated with higher scores within the four ADV subscales, while being a school aggressor was only associated with having perpetrated physical/sexual or psychological ADV.

Table 4 shows the Cohen’s values of the distribution of ADV subscales across the variables, separately from boys and girls. Overall, the strongest associations to dating violence were found within the relationship power imbalance items, suggesting that these items may also be useful to detect dating violence situations. Among boys, all items except those related to substance use were associated with at least one form of dating violence. Among girls, substance use was related to both suffered and perpetrated violence, and fewer relations were found between opinions related to sexual violence in relationships and dating violence outcomes.

## 4. Discussion

The ADV-YL questionnaire seems to be a valid and reliable instrument to assess adolescent dating violence in Spanish-speaking countries. It includes items related to psychological, physical, and sexual violence, both suffered and perpetrated, and taking place in person or virtually.

Regarding ADV-YL’s factor structure, our results show two different constructs: psychological violence and physical/sexual violence. In both perpetrated and suffered violence, items consistently measure the same construct. The resulting structure is coherent with the conceptual approach, and similar to the original proposed model of psychological/physical/sexual violence. Specifically, we obtained two factors, one measuring psychological ADV and another measuring physical/sexual ADV. Furthermore, the ADV subscales exhibit good internal consistency. In addition, the structure of the questionnaire was confirmed in the three countries through the Confirmatory Factor Analyses.

The highest means were found for the suffered psychological violence scale, followed by perpetrated psychological violence. Physical/sexual violence yielded the lowest scores. These findings are consistent with scientific research about ADV showing that psychological violence is more frequent among adolescents than physical or sexual violence [3,4]. Although the prevalence of dating violence in this study was low, the ADV-YL tool seems to adequately distinguish different forms of dating violence.

There were also differences between suffered and perpetrated ADV, with higher scores for the former. This may be the result of social desirability bias. Adolescents may prefer to conceal both types of events, but this effect might be stronger for perpetrated violence. A meta-analytic review about couples’ agreement for physical aggression showed that both sexes underreported their own aggression more frequently than their victimization [39]. Another study among young people found that women and men more easily identified controlling behaviors in other partners than in their own relationships. In addition, men reported that such behaviors were common among young couples, but that they never or rarely controlled their partners [40].

We found moderate correlations between all types of suffered and perpetrated ADV. This confirms the specific pattern of dating violence during adolescence, where reciprocal aggression seems to be common [4,10,11,12,13]. This finding has important implications. If a female or a male adolescent is suffering dating violence, we should also explore if they are also perpetrating it pro-actively, or as a response to the aggression received. In clinical settings, exploring whether dating violence is uni- or bidirectional is important to correctly plan the therapeutic approach. Additionally, in school settings, knowing whether students are suffering and/or perpetrating dating violence could help to develop more specific prevention programs. It has to be highlighted that all types of dating violence need to be avoided in order to prevent their consequences and other aggressive behaviors, as it is well-known that violence leads to violence [41].

We also analyzed the associations between several variables, expected to covariate with ADV, and the subscales of ADV. ADV scores were higher among those reporting the use of substances. When separating by sex, these associations persisted mainly for girls while, for boys, only the use of “other drugs” (cocaine, amphetamines, ecstasy, etc.) was significantly associated with ADV. Drug consumption is a well-known risk factor for both perpetrating and suffering intimate partner violence [2], although more evidence is needed for some drugs [42]. This is not surprising. The psychopharmacological effect of alcohol and other drugs is well-known. They may affect cognitive processes and impair impulse control [43]. Being drunk or drugged may jeopardize sexual consent, leading to sexual assault even among dating partners. On the other hand, according to the Problem-Behavior Theory [44], alcohol and drug abuse usually occurs together with other risk behaviors, such as violence, although there may not necessarily be a causal relationship between them. Regardless of whether the association is causal, studies show that adolescents who consume alcohol or drugs are more vulnerable to intimate partner violence [21,45,46]. Therefore, it is important to take this into account when designing preventive programs for adolescents. Prevention of bullying or alcohol consumption could also contribute to the prevention of dating violence.

We also found that students suffering or perpetrating school aggression obtained higher scores for psychological, sexual, and physical dating violence than those who did not report it. These results were different for boys and girls. For boys, having suffered any school aggression was associated with psychological ADV subscales (both suffered and perpetrated). For girls, the significant associations were between having perpetrated school aggression and having perpetrated ADV. According to the scientific literature, bullying and perpetrating violence are usually linked. On the one hand, studies show that school bullying is a predictor for ADV and adult intimate partner violence [47,48]. As explained by Ellis and Wolfe, “the use of patterns of aggressive behavior among peers leads to increased status and social recognition for those who employ them, and are therefore internalized as a valid form of relationship and used in other types of relationships” [49]. On the other hand, bullying and dating violence may share risk factors such as lack of empathy, impulsivity, low family cohesion, depressed affect, or anger reactivity, among others [16]. These findings point out that bullying prevention may contribute to ADV prevention.

It was not surprising that the four ADV subscales were associated with all items about the justification of violence in relationships. When analyzed separately for boys and girls, all the associations remained strong and significant for boys. For girls, justification of physical violence (but not justification of sexual violence) and relationship power imbalance were associated with all the suffered and perpetrated ADV subscales. It is important to identify students who justify violence behaviors, as they are at great risk of suffering or perpetrating dating violence [50,51].

As expected, the four ADV subscales were also associated with the three items about relationship power imbalance. These items could be considered an indirect measure of ADV. This was also found in a research conducted among adult couples planning their wedding [23]. Therefore, power imbalance items could be used for ADV screening, when long versions of ADV questionnaires cannot be implemented, or when circumstances (such as ethical or legal issues) make it difficult to explicitly ask about violence.

An essential step in the prevention of ADV is to identify and quantify its existence. The ADV-YL tool can be easily implemented in schools to detect the situations of dating violence among their students. Sharing the results with the students could help raise awareness of this serious problem and make it easier to seek help for those involved in dating violence.

The thorough study of ADV and its determinants needs a comprehensive list of variables and thus long questionnaires. Such complicated studies are not easily feasible. Fortunately, the ADV-YL tool has shown to be useful, among Hispanic populations, to study and assess the determinants of dating violence.

Future studies are needed to evaluate the validity of the scale in other populations from different cultural settings.

Some limitations of this analysis should be noted. First, public and non-confessional schools were underrepresented in the sample. It has been suggested that religiosity is a protective factor for several risky behaviors or negative outcomes [52,53]. Therefore, students in our sample might be classified as a low risk violence group and high homogeneity in responses would be expected. In spite of this, the questionnaire has adequately distinguished between different types of dating violence. Second, we did not include other instruments to calculate concurrent and discriminant validity. However, some of the predictive variables were proxies of the construct (i.e., variables about relationship power imbalance or justification of violence). Therefore, they could be used to assess concurrent validity, to some extent. They were expected to correlate with ADV scale and, in fact, they did so (Table 3 and Table 4). Third, the ADV instrument does not measure whether perpetrated violence is reactive to suffering violence. It has to be noted that our instrument was developed to assess risks and protective factors for any type of ADV among schooled adolescents within the YourLife project. For this purpose, the four subscales of violence assessed by this questionnaire would be adequate. Finally, evaluation of the test–retest reliability would be needed once any longitudinal follow-up is to be conducted. Forth, we successfully tested configural invariance, but not other types of invariance. However, we did not intend to study inter-culture differences; we only aimed to test whether the instrument is valid in the three countries, and with the same factor structure. In any case, future studies may investigate this issue further.

## 5. Conclusions

In conclusion, our instrument seems to be adequate to assess ADV within a multipurpose questionnaire. It identifies suffered and perpetrated ADV, including psychological, sexual, and physical violent behaviors, in person or virtually, among schooled adolescents. Items related to relationship power imbalance might also be used to detect ADV when a more direct questionnaire cannot be implemented.

Correlations between all types of suffered and perpetrated dating violence were found, showing that bidirectional violence is common among adolescents. In addition, ADV was also associated to other risk behaviors such as substance use or bullying. These findings could help to develop more specific prevention programs.

## Figures and Tables

**Figure 1 ijerph-18-06824-f001:**
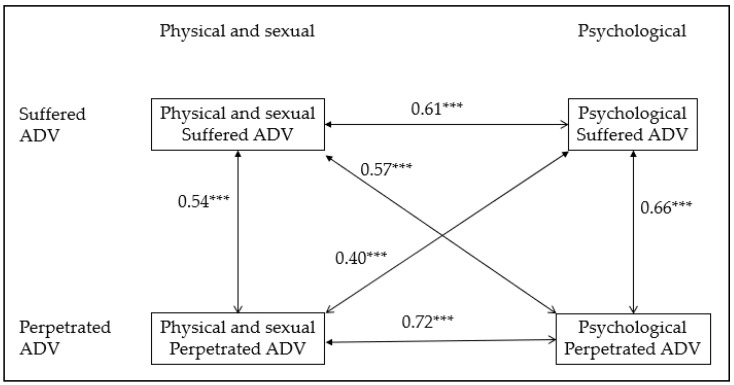
Pearson coefficients for the correlations among Adolescent Dating Violence (ADV) Subscales. *** All correlations were statistically significant (*p* < 0.001).

**Table 1 ijerph-18-06824-t001:** Matrix with factor loadings from principal component analysis with Promax rotation.

Item		% Ever ^1^	Factor 1 Physical/Sexual Violence	Factor 2 Psychological Violence	Uniqueness
**Suffered violence ^2^**					
01	Your partner insults, criticizes, or shouts at you (in person, over the phone, on social media)	24.5		0.5898	0.5650
02	Your partner does not allow you to speak to or see your family or friends	12.8		0.6013	0.5879
03	Your partner calls or texts you constantly to control what you’re doing, where you are, or who you’re with	26.7		0.7460	0.4883
04	Your partner threatens to leave you when you fight	12.5		0.5923	0.6008
05 *	Your partner threatens to hurt you if you leave them	5.8	0.4188		0.5226
06	Your partner sets the rules (schedules, dates, etc.) for your relationship without considering your opinion	11.8		0.6447	0.4580
07	Your partner has checked your phone without permission	18.1		0.7550	0.4883
08	Your partner controls what you do on social media	20.4		0.8100	0.4211
09	Your partner forces you to perform sexual acts	5.3	0.8414		0.2857
10	Your partner touches parts of your body sexually although you said you did not want him/her to	10.6	0.6009		0.5257
11	Your partner says that he/she will leave you if you do not have sexual relations or perform other types of sexual acts with him/her	4.5	0.8181		0.3777
12	Your partner takes advantage of the fact that you have had alcohol or drugs to have sexual relations with you or perform any other sort of sexual act with you	5.9	0.7665		0.4678
13	Your partner forces you to send erotic/sexual photos or videos of yourself	5.6	0.8758		0.3807
14	Your partner sends you erotic/sexual photos or videos of him/herself, even though you said you did not want to receive them	7.6	0.6624		0.5355
15	Your partner has threatened to hurt you, harm, you or throw something at you	4.9	0.6697		0.4430
16	Your partner has hit you, kicked you, pulled your hair, or thrown something at you	6.8	0.5627		0.4772
17	Your partner has grabbed or pushed you forcefully	9.4	0.4889		0.4243
18	Your partner ruins or threatens to ruin things that you really like	5.0	0.6769		0.3878
**Perpetrated violence ^3^**					
01	I insult, criticize or shout at my partner (in person, over the phone, on social media)	15.2		0.6874	0.4988
02	I do not allow my partner to speak to or see his/her family or friends	5.4		0.4172	0.4868
03	I call or text my partner constantly to control what he/she is doing, where he/she is, or who he/she is with	16.2		0.7327	0.5007
04	I threaten to leave my partner when we fight	6.7		0.6014	0.4707
05 *	I threaten to hurt my partner if he/she leaves me	2.8	0.6433		0.3305
06	I set the rules (schedules, dates, etc.) for our relationship without considering my partner’s opinion	6.3		0.5509	0.3625
07	I have checked my partner’s phone without permission	12.9		0.8037	0.4062
08	I control what my partner does on social media	15.4		0.707	0.5432
09	I force my partner to perform sexual acts	3.1	0.9831		0.1297
10	I touch parts of my partner’s body sexually although he/she said did not want me to	5.2	0.8449		0.2546
11	I say that I will leave him/her if my partner does not have sexual relations or perform other types of sexual acts with me	2.9	0.962		0.1396
12	I take advantage of the fact that my partner has had alcohol or drugs to have sexual relations with him/her or perform any other sort of sexual act with him/her	3.8	0.7513		0.3281
13	I force my partner to send erotic/sexual photos or videos of himself/herself	3.0	0.9603		0.0969
14	I send my partner erotic/sexual photos or videos of myself, even though he/she said he/she did not want to receive them	3.3	0.9085		0.1294
15	I have threatened to hurt, harm, or throw something at my partner	2.7	0.8642		0.1693
16	I have hit or kicked my partner, pulled his/her hair, or thrown something at him/her	4.1	0.706		0.3038
17	I have grabbed or pushed my partner forcefully	6.1	0.6316		0.3027
18	I ruin or threaten to ruin things that my partner really likes	3.0	0.8688		0.1148

Note: Weights below 0.4 are omitted. ^1^ Percent of participants who gave responses different to “Never”. ^2^ Questions used to measure suffered violence in present or past romantic relationships ^3^ Questions used to measure perpetrated violence in present or past romantic relationships * This item was excluded in the final version of the questionnaire as it had both psychological and physical qualities.

**Table 2 ijerph-18-06824-t002:** Fit indices in Confirmatory Factor Analysis in each country.

Adolescent Dating Violence (ADV)	Sample	χ^2^	df ^a^	χ^2^/df	RMSEA (90%CI) ^b^	CFI ^c^	SRMR ^d^
Suffered ADV	Spain	95.517	67	1.426	0.029 (0.014–0.041)	0.996	0.021
	Chile	75.507	67	1.127	0.019 (0.000–0.038)	0.998	0.036
	Ecuador	57.165	67	0.853	0.000 (0.000–0.030)	1.000	0.036
Perpetrated ADV	Spain	101.131	67	1.509	0.031 (0.018–0.043)	0.998	0.010
	Chile	174.703	67	2.608	0.068 (0.056–0.081)	0.980	0.067
	Ecuador	89.762	67	1.340	0.043 (0.013–0.065)	0.993	0.029

^a^ Degrees of freedom. ^b^ Root Mean Square Error of Approximation (and 90% confidence interval). ^c^ Comparative Fit Index. ^d^ Standardized Root Mean squared Residual.

**Table 3 ijerph-18-06824-t003:** Differences in the adolescent dating violence (ADV) subscales across groups defined by some predictor variables.

			Suffered				Perpetrated			
			Psychological ADV		Physical/Sexual ADV		Psychological ADV		Physical/Sexual ADV	
		n	Mean ^a^	*d* ^b^	Mean ^a^	*d* ^b^	Mean ^a^	*d* ^b^	Mean ^a^	*d* ^b^
**Demographics**										
Sex	Male	482	0.51	−0.05	0.21	−0.11	0.24	0.07	0.12	−0.10
	Female	567	0.47		0.14		0.29		0.07	
Age	13–15	436	0.39	0.18 **	0.16	0.04	0.22	0.13 *	0.10	−0.02
	16–18	613	0.56		0.18		0.31		0.09	
Type of school	Co-educational	931	0.51	−0.20 *	0.17	−0.01	0.28	−0.16	0.10	−0.10
	Single-sex	118	0.32		0.17		0.17		0.05	
Area of school	Rural	87	0.46	0.03	0.18	−0.02	0.26	0.01	0.14	−0.09
	Urban	962	0.49		0.17		0.27		0.09	
**Substance use ^c^**										
Alcohol	No	311	0.39	0.16 *	0.14	0.07	0.25	0.04	0.10	−0.03
	Yes	717	0.54		0.19		0.28		0.09	
Binge drinking	No	692	0.44	0.16 *	0.14	0.18 **	0.23	0.16 *	0.07	0.13 *
	Yes	342	0.59		0.25		0.35		0.14	
Cannabis	No	866	0.45	0.24 **	0.15	0.22 **	0.24	0.24 **	0.08	0.16
	Yes	169	0.68		0.29		0.42		0.16	
Other drugs	No	1010	0.46	1.24 ***	0.14	2.00 ***	0.25	1.46 ***	0.07	1.63 ***
	Yes	26	1.59		1.28		1.27		0.89	
**School peer aggression**										
Suffered	No	504	0.41	0.16 **	0.13	0.15 *	0.20	0.18 **	0.06	0.12 *
	Yes	526	0.56		0.22		0.33		0.13	
Perpetrated	No	582	0.45	0.10	0.15	0.10	0.21	0.19 **	0.06	0.15 *
	Yes	452	0.54		0.20		0.34		0.14	
**Justification of violence**										
Hit your partner ^d^	No	934	0.42	0.79 ***	0.11	1.10 ***	0.21	0.94 ***	0.04	1.07 ***
	Yes	101	1.13		0.74		0.85		0.57	
Forced sex ^e^	No	908	0.44	0.43 ***	0.12	0.78 ***	0.22	0.61 ***	0.05	0.73 ***
	Yes	121	0.84		0.58		0.65		0.42	
Sex when drunk ^f^	No	899	0.45	0.37 ***	0.12	0.72 ***	0.23	0.48 ***	0.05	0.65 ***
	Yes	131	0.79		0.55		0.56		0.38	
**Relationship power imbalance ^g^**										
Frightened/afraid	No	838	0.33	0.93 ***	0.10	0.61 ***	0.20	0.47 ***	0.06	0.28 ***
	Yes	205	1.13		0.45		0.53		0.20	
Trapped	No	682	0.28	0.68 ***	0.10	0.35 ***	0.19	0.31 ***	0.07	0.13 *
	Yes	360	0.88		0.30		0.41		0.13	
Controlled	No	759	0.24	1.07 ***	0.10	0.43 ***	0.20	0.37 ***	0.07	0.12
	Yes	283	1.14		0.36		0.46		0.13	

^a^ Mean of subscale scores within that category. ^b^ Cohen’s *d* of the difference of means of the ADV subscale between the two groups defined by the predictor variables. ^c^ Ever consumed in the past 12 months. Binge drinking: 4–5 alcoholic drinks in few hours. Other drugs: cocaine, amphetamines, ecstasy, etc. ^d^ Any degree of agreement with the sentence “Sometimes it is justifiable to hit your partner if the other person does something annoying”. ^e^ Any degree of agreement with the sentence “Sometimes it is justifiable to have sex although the other person does not want to”. ^f^ Any degree of agreement with the sentence “Sometimes it is justifiable to have sex when the other person is drunk”. ^g^ Any frequency of having had the indicated feeling toward the partner. *p*-value for the Student’s *t*-test: * *p* < 0.05, ** *p* < 0.01, *** *p* < 0.001.

**Table 4 ijerph-18-06824-t004:** Differences in the adolescent dating violence (ADV) subscales across groups defined by some predictor variables, stratified by sex.

	Boys					Girls				
	% ^1^	Cohen’s *d* ^2^				% ^1^	Cohen’s *d* ^2^			
		Suffered		Perpetrated			Suffered		Perpetrated	
		Psy	Phys/ Sex	Psy	Phys/ Sex		Psy	Phys/ Sex	Psy	Phys/ Sex
**Demographics**										
Age: 16–18 years (ref. = 13–15 years)	55.8	0.20 *	0.06	0.15	0.05	60.7	0.17 *	0.04	0.11	−0.09
Type of school: Single-sex (ref.: co-educational)	11.8	−0.23	−0.05	−0.16	−0.15	10.8	−0.18	0.03	−0.17	−0.05
Area: Urban (ref. = Rural)	91.9	−0.13	−0.14	−0.08	−0.18	91.5	0.16	0.11	0.08	0.00
**Substance use ^3^**										
Alcohol consumption	64.2	0.12	−0.01	−0.06	−0.09	74.5	0.21 *	0.22 *	0.14	0.08
Binge drinking	33.5	0.15	0.11	0.04	0.00	32.7	0.16	0.26 **	0.26 **	0.27 **
Cannabis	17.0	0.18	0.18	0.04	0.00	15.8	0.30 *	0.27 *	0.43 ***	0.33 **
Other drugs	2.9	1.23 ***	1.91 ***	1.37 ***	1.20 ***	2.1	1.24 ***	2.07 ***	1.58 ***	2.16 ***
**School peer aggression**										
Suffered	48.5	0.20 *	0.15	0.24 **	0.18	53.2	0.14	0.16	0.13	0.08
Perpetrated	51.7	0.04	0.07	0.16	0.08	37.0	0.13	0.10	0.24 **	0.20 *
**Justification of violence**										
Hit your partner ^4^	12.2	0.88 ***	1.51 ***	1.26 ***	1.46 ***	7.7	0.68 ***	0.47 **	0.63 ***	0.54 ***
Forced sex ^5^	19.5	0.60 ***	1.03 ***	0.89 ***	0.96 ***	5.2	0.07	−0.14	0.19	0.08
Sex when drunk ^6^	21.2	0.52 ***	0.91 ***	0.73 ***	0.83 ***	5.5	0.04	−0.01	0.09	0.09
**Relationship power imbalance ^7^**										
Frightened/afraid	21.2	0.67 ***	0.54 ***	0.68 ***	0.56 ***	18.4	1.19 ***	0.69 ***	0.30 **	−0.01
Trapped, unable to leave the relationship	37.1	0.61 ***	0.35 ***	0.43 ***	0.32 ***	32.4	0.74 ***	0.33 ***	0.22 *	−0.06
Controlled, lacking freedom	30.6	1.00 ***	0.36 ***	0.42 ***	0.26 *	24.2	1.15 ***	0.52 ***	0.35 ***	−0.04

Psycho: Psychological adolescent dating violence, Physic/Sex: physical and sexual adolescent dating violence. Ref.: reference. ^1^ Percentage of participants in the category indicated in the left column (the denominator is the number of boys or girls). ^2^ Cohen’s *d* of the difference of means of the ADV subscale between the two groups defined by the predictor variables. ^3^ Ever consumed in the past 12 months. Binge drinking: 4–5 alcoholic drinks in few hours. Other drugs: cocaine, amphetamines, ecstasy, etc. ^4^ Any degree of agreement with the sentence “Sometimes it is justifiable to hit your partner if they do something annoying”. ^5^ Any degree of agreement with the sentence “Sometimes it is justifiable to have sex although they do not want to”. ^6^ Any degree of agreement with the sentence “Sometimes it is justifiable to have sex when they are drunk”. ^7^ Any frequency of having had the indicated feeling toward the partner. * *p* value for Student’s *t*-test: *p* < 0.05, ** *p* < 0.01, *** *p* < 0.001.

## Data Availability

The data presented in this study are available on request from the corresponding author. The data are not publicly available as they are still being exploited.

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
