# Peer review of "Assessing Adolescent Dating Violence in the YourLife Project: Proposal of an Instrument for Spanish-Speaking Countries"

_ijerph, 2021, doi:10.3390/ijerph18136824_

Round 1

Reviewer 1 Report

Thank you for submitting your manuscript. The topic is relevant and the paper is well written. 

Page 1

Line 11: Please remove the acronym in the abstract.

Page 2

Line 47-48: Explain why alcohol consumption or bullying can lead to dating violence.

Line 90-91: Indicate the mean age of girls and boys who participate in the study.

Page 11

Line 321-325: Conclusion should be more extensive.

Reviewer 2 Report

First of all, I would like to congratulate the authors for their work. It is really very well thought out, it addresses a field that needs to be updated and studied. Moreover, it includes information that can be very valuable for the scientific community in general, and especially for psychology. I think that the work could be published with some modifications, mainly suggestions that imply the inclusion of more information and the clarification of different points.

1. Title, abstract and introduction.

The title seems to me adequate and illustrative
The introduction is very well focused but should address all the variables that are later analysed, please include these aspects.
It is important that you contextualise in the introduction the possible cultural differences in this respect in the different countries. It is possible that if the instrument does not show adequate invariance it is because of this.

2. Method.
They should include the percentage of adolescents from each country, as well as the type of school they came from.
A major limitation is that they have not included other instrument(s) to calculate concurrent and discriminant validity. Is it possible to remedy this?
Were data collected during the pandemic? If not, please clarify this. Collecting data at both points in time may have biased the data.
Extend information about ethical considerations (signing of informed consent, ethics committee, etc.). 

3. Results.
Have you tried to test the invariance of the instrument?
You can calculate the composite reliability of the instrument, this procedure is strongly recommended.
Some outcome variables are qualitative, it is incorrect to use a t-test in comparisons where both are qualitative. Use chi-square and cramer's v.
I do not understand table 4, as the statistic used is not reported and different types of variables are mixed.
4. Discussion.
I think the limitations can be extended (design, analysis...).
What are the implications of your study? What are the future lines of research?

5.  OTHER ASPECTS
The figure should be in the same typeface as the rest of the manuscript.
Again, I thank the authors for their effort and work.

Reviewer 3 Report

Congratulations on your work; I am impressed with your thoroughness.  I do have some specific suggestions, however, that I would want to see implemented before accepting the submission.

  1. Lines 24-45: One of the reasons this paper is valuable is because it is specific to Hispanic countries.  For that reason, please clarify the populations addressed by the studies given in the introduction.  For example, which populations are covered by the systematic review paper [2]? Reference [4] appears to be specific to Danish seventh-graders; [5] appears to be specific to Spanish HS students. Please rewrite to be clear when the reference, and therefore the trend or characteristic being highlighted, is specific to a particular population or relevant across multiple populations.  
  2. Line 53: Replace "is" with "can be" to reflect the fact that the reference material is specific to a particular population.
  3. Line 65: it appears there is a grammatical error; should "their" be "its"?
  4. Line 77-79: Combine these two sentences with a semi-colon as the second sentence, while long, does not appear to have a subject-verb structure.
  5. Line 80: "ad hoc" is typically a little less deliberate than what you did!  I would replace "ad hoc" with "newer" or "abbreviated".
  6. Lines 87-91: I am not able to determine from the information you have provided was the nonresponse level is. The original sample is 2254. How many respondents were there in the 13-18 year-old group? How many of those reported experience with dating?  How many of those were dropped due to missing data?  Your final sample is 1,049; that is a relatively small fraction of 2,254 but could be the majority of the dating 13-18 year olds; I want to know so I can judge for myself whether there might be a nonresponse bias issue.
  7. Line 104: Table 1's placement is a bit awkward; to address that, I recommend adding the sentence "A description of the factor analysis results presented in Table 1 follows in Section 2.4." at line 104.
  8. Lines 150-168: My biggest concern related to your submission is the very small section devoted to explaining the methodology for the data analysis, but this is easily fixable. There are no references here, not even to the software package you chose to use.  Since this journal is interdisciplinary in its appeal, I strongly believe you should at least provide enough references so that a reader unfamiliar with factor analysis and measures of validity will be able to learn more. If you used SPSS for your analysis, the appropriate SPSS manual could be used as a single reference.
  9. Line 199: p<0.001 (you are missing a zero).
  10. Line 220: The indenting in the demographics section needs to be fixed.
  11. Line 227: ref. should be described in the footnotes for the table like all the other abbreviations.
  12. TABLES: There are no descriptive statistics given for the basic data. This is an issue in Tables 1, 3 and 4.  In Table 1, the original "averages" are not given for the questions (inasmuch as data from a Likert scale from 0 to 6 can have a measure of center given through an average). In Table 3, overall means across the sample for each characteristic are not given; for example, what proportion of the sample overall said yes to binge drinking? At the very least, this allows us to determine if any of these categories have unusually small sample sizes. Alternatively, an overall number of participants with the characteristics (n=) could be given for each row. Table 4 has similar issues.

Round 2

Reviewer 2 Report

First of all, I would like to thank the authors for their considerations and changes. I think they have done an important job in improving the manuscript. 

Regarding the invariance of the instrument, I understand that they do not want to carry out this analysis although it is relevant. In fact, if it did not "come out" right, it should be reported that the instrument does not work separately. Please consider this aspect for future work. However, since you have not addressed it, I suggest you add this as a limitation/future line of research. 

Thank you for clarifying information about this comment "9) Some outcome variables are qualitative, it is incorrect to use a t-test in comparisons where both are qualitative. Use chi-square and cramer's v.
In Tables 3 and 4, the independent variables (left column) are dichotomous. The dependent variables (the four ADV subscales, named in the column headers) are continuous". I think that, as you point out, the headers need to be changed to make it clearer. Because the title talks about "associations" and they look more like differences. 
As for table 4, it would be appropriate to put the names of the statisticians in the relevant columns to make it understandable. In addition, this table should be in the same font as the manuscript (it seems that the content is in calibri).

Thank you again for your work and kindness,
With best wishes,
